# Veracity: An Online, Open-Source Fact-Checking Solution

## Abstract

The proliferation of misinformation poses a significant threat to society, exacerbated by the capabilities of generative AI. This demo paper introduces Veracity, an open-source AI system designed to empower individuals to combat misinformation through transparent and accessible fact-checking. Veracity leverages the synergy between Large Language Models (LLMs) and web retrieval agents to analyze user-submitted claims and provide grounded veracity assessments with intuitive explanations. Key features include multilingual support, numerical scoring of claim veracity, and an interactive interface inspired by familiar messaging applications. This paper will showcase Veracity's ability to not only detect misinformation but also explain its reasoning, fostering media literacy and promoting a more informed society.

## 1 Introduction

Experts have rated the dissemination of misinformation and disinformation as the #1 risk the world faces Torkington (2024). This risk has only increased with the proliferation and advancement of generative AI Bowen et al. (2024); Pelrine et al. (2023b). Responses to misinformation have up to now been largely limited to platform moderation. As large-scale social media platforms actively eliminate their content moderation teams and shrug off their social responsibility in preventing the manipulation of their users Horvath et al. (2025), fact-checking and misinformation detection are being forced onto the end user. In the absence of strong platform-based approaches, solutions that support individuals with fact-checking tools become essential in dampening the societally corrosive effects of misinformation.

Misinformation is particularly dangerous when it influences public health and democratic processes, as seen in the spread of vaccine-related disinformation and politically motivated claims about censorship, both of which have been shown to exacerbate real-world harm and undermine trust in institutions Lewandowsky (2025). With the rollback of content moderation efforts and increasing concerns over algorithmic bias on social media platforms, independent, reliable fact-checking tools are more necessary than ever.

A promising solution in this area is an AI Steward that helps people fact-check and filter out manipulative and fake information. In fact, AI can outperform human fact-checkers in both accuracy Wei et al. (2024); Zhou et al. (2024) and helpfulness Zhou et al. (2024). Although there is rapid progress in improving the accuracy of such systems Tian et al. (2024); Wei et al. (2024); Ram et al. (2024), there is much less research on how to make a high accuracy system into a helpful and trustworthy one that users can rely on. Our AI-powered open-source solution, **Veracity**, deploys large language models (LLMs) working in conjunction with web retrieval agents to provide any member of the public with efficient and grounded analysis of the veracity of user-inputted text.

### 1.1 Problem Scenario

Our society needs tools that support information integrity by defending against rampant misinformation. Individuals currently face the challenge of combatting disinformation largely on their own. Individuals face a lack of 'good' information, and also difficulty in reliably finding information from credible sources to justify the whether or not a statement in question is true or false. Tools that

help individual users address this challenge exist, but are either proprietary, in which case there are access, transparency and privacy issues, or are limited in their ease of use.

## 1.2 PROPOSED SOLUTION

We propose a fact-checking system solution that uses a Large Language Model (LLM) to summarize relevant text retrieved by a web agent from reliable sources on the internet. The solution was designed to address the following goals related to information integrity:

- Counter misinformation by providing accurate, evidence-based assessments.
- Foster media literacy by helping users critically evaluate online claims.
- Promote transparency by explaining why a claim is assessed as true or false.
- Ensure broad accessibility, making fact-checking tools open-source and available to anyone.

The system was designed for:

- The general public, including both tech-savvy users and those less familiar with new technologies.
- Expert users, such as journalists and professional fact-checkers who require efficient and reliable verification tools.

## 1.3 OUR CONTRIBUTION

This paper describes the design and functionality of an open-source, claim-focused fact-checking system that is designed to enhance transparency in model decision-making. We detail its application domain, technical architecture, AI techniques, and interactive elements. Unlike traditional black-box models, our applications allows users to submit claims and receive structured responses that provide clear analysis on how reasoning was done to reach the veracity decisions. With a strong emphasis on open research, our system is build to be fully accessible to anyone, so anyone can download the application and run it locally. This is important for ensuring reproducibility and collaboration/feedback from the community. Key features include multilingual support, a numerical scoring for claim veracity, and we also demonstrate how this tool addresses misinformation by developing an intuitive, transparent platform for claim verification.

## 2 SYSTEM DESCRIPTION

### 2.1 SYSTEM OVERVIEW

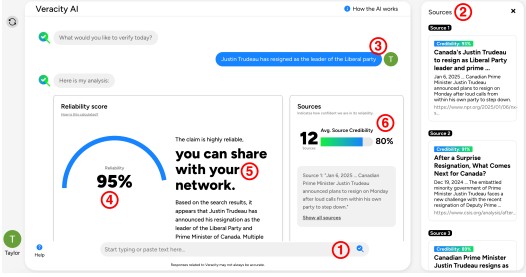

Figure 1: The main fact-checking page of Veracity

The main functionality of the system can be seen in Figure 1. This is the system's main page, where the user is taken immediately upon logging in. The behavior of each part of the interface is described by the numerical mappings shown in Figure 1:

1. **Claim submission box**: This box is where the user can type or copy/paste the claim they want the AI to verify.

2. **Sources panel**: When a claim is submitted, the LLM will (if it decides it is necessary) use a web-agent to retrieve sources, all of the sources used to evaluate a claim will be displayed here.

3. **Claim under analysis:** After a user submits a claim through the claim submission box, it is displayed on the screen.

4. **Reliability score:** This is the score generated by the LLM that reflects the reliability of the claim, where 0% maps to completely unreliable or false and 100% maps to completely reliable or true.

5. **Textual instruction per reliability score & LLM explanation**: The user is shown an actionable message that interprets the model's veracity score (0-20% is mapped to the claim is not reliable, 21-40% is mapped to the claim is likely not reliable, 41-60% is mapped to the claim needs further investigation, 61-80% is mapped to the claim is reliable, 81-100% is mapped to the claim is highly reliable) and a share recommendation (the score must be greater than 60% for a positive recommendation). Below this is the LLM reasoning that explains its reliability score.

6. **Source summary**: This includes aggregate information about the sources used to determine the reliability of the claim, including the number of sources and the average credibility ranking of the sources.

## 2.2 TECHNOLOGY STACK

The system is divided into separate frontend and backend tech stacks, with the frontend being served by HTTPS requests to an application programmable interface (API). The frontend and backend exist separately, except for the API that forms a contract between the two.

### 2.2.1 FRONTEND

The web display, or visualization of the application, was implemented using the Next.js Vercel (2025) and deployed using the Vercel deployment pipeline within the package. The frontend also uses Sass, Typescript, and Chart.js Chart.js Contributors (2025). For complete documentation on the frontend technology stack, please see the frontend project wiki [link].

### 2.2.2 BACKEND

The backend, encompassing the application logic and the persistence (i.e. database) layers is deployed using the Google Cloud Platform (GCP). The application logic or API is deployed on Kubernetes, and the database is deployed on Cloud SQL Cloud (2025b;a). Beyond deployment, the API is designed using FastAPI Ramírez (2025), the database is implemented in PostgreSQL Group (2025), and the object mapping between the API and the database is managed by SQLAlchemy Bayer (2025). For full documentation on the backend technology stack, please see the backend project wiki [link].

## 3 AI TECHNIQUES AND INNOVATIONS

### 3.1 CORE AI METHODS

This system uses AI to power its fact-checking methodology, specifically LLM technology. Despite the challenges of misinformation detection, including the tendency of misinformation to contain a mix of both true and false information, LLMs have been shown to be effective tools for detecting misinformation online Pelrine et al. (2023a); Chen & Shu (2024). However, LLMs alone may not be enough. Many studies have shown the benefits of retrieving information from online sources to improve the performance of fact-checking and misinformation detection Bekoulis et al. (2021); Kondamudi et al. (2023); Zhou & Zafarani (2020).

To achieve this goal, the system is an implementation of the LLM/web search engine teaming proposed by Tian et al. [2024] in their *Web Retrieval Agents for Evidence-Based Misinformation Detection*. The interactions between the LLM, web search engine, and user are described by Figure 2.

## 3.2 INNOVATIONS

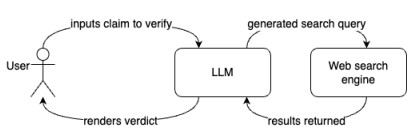

Figure 2: The interactions between the LLM, web search engine, and user

This system represents a unique innovation and application of AI in the fact-checking space. In addition to the unique teaming of web search agents and LLM reasoning as described above, this system has a couple of important innovations or distinctions from other AI fact-checking systems. In particular, this is due to a few important features:

- **Sources display**: Not only does the system use a search engine to select relevant sources, the LLM is shown these sources to help it render its verdict. The user is also shown a list of sources, as well as their documented credibility Lin et al. (2023).

- **Score-based analysis**: This is the first tool of its kind to present for a reliability score that represents the factuality of a user's claim and to ask the LLM to justify its presentation of this score.

## 4 INTERACTIVE ELEMENTS

The system was designed to invoke a feeling of familiarity and trust from all users, while prioritizing the interactivity of the system. The modalities of interaction, as well as the central interface, were inspired by standard messaging applications (WhatsApp, Messenger, etc.). In addition to the main interaction (a user submits a claim and reviews the result), there are a few extra interactive elements that are discussed below.

### 4.1 COLLECTION OF USER FEEDBACK

The system is designed to enable continuous improvement. This is done by user feedback. The feedback mechanism is user-driven and is specific to the system's analysis of a particular claim. The user can select a rating between 1 and 5 stars to reflect how well the model analyzed their claim. Following this selection, the user can select a series of 'tags' or small textual snippets that reflect different functionalities of the system, for example, the sources. They may also submit an optional comment.

### 4.2 EXPERT DASHBOARD

The system is also unique in that it is not just designed for users to fact-check relevant claims. Registered users who identify themselves as experts, and are approved as such by the system administration, will have access to a fact-checking expert dashboard. This dashboard is designed to display aggregate information from the application to these users. For example, it displays a clustering graph which captures the most common trends in claims submitted to the system.

## 5 CONCLUSION

This demo has showcased Veracity, an open-source AI system that combines LLMs and web retrieval agents to provide transparent and accessible fact-checking. While AI systems employing LLMs and web retrieval for fact-checking exist, open-source versions are not readily available. Veracity aims to fill this gap by providing production ready veracity assessment application, with intuitive explanations, i) empowering individuals to critically evaluate information and contribute to a more informed society; ii) empowering the research community to expand the system's capabilities and build the next generation of AI-powered fact-checking systems. Future work include improving handling of complex claims, enhancing user interaction features, broadening the language and context support, and the integration of more advanced credibility measurement techniques. Veracity's open-source nature encourages community involvement and further development to address the ongoing challenge of misinformation.

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

Supplementary Materials Below are important links to learn more about the system:

- Frontend GitHub repository
- Backend GitHub repository
- Beta Deployment

