# OpenReview forum: "Veracity: An Online, Open-Source Fact-Checking Solution"
_ICLR.cc/2025/Workshop/BuildingTrust — Submitted to BuildingTrust_

### Official Review · Reviewer_6yid · 2025-02-27
**Veracity: An Online, Open-Source Fact-Checking Solution**

**Rating:** 5
**Confidence:** 3

**Review:**

The combination of Large Language Models (LLMs) and information retrieval presents a promising approach to enhancing reliability in fact-checking. While the proposed approach is valuable for LLM applications, greater transparency is needed regarding the factors LLMs consider when assigning reliability scores. While the paper mentions credibility assessments for sources, it does not explain how these scores are calculated—a more detailed discussion would improve clarity. Additionally, further information on the LLM itself, including its pre-training data size and source (whether it has been self-developed or used open source LLM), would strengthen the paper’s technical foundation. Providing actual examples illustrating the model’s accuracy in distinguishing facts would reinforce its practical effectiveness. Similarly, clarifying whether the information retrieval method is compatible with various retrieval techniques would enhance the paper’s scope. Finally, empirical experiments demonstrating the method’s performance would significantly improve the study’s credibility.

---

### Decision · Program_Chairs · 2025-03-04

Reject